# SPREADING VECTORS FOR SIMILARITY SEARCH

Alexandre Sablayrolles[†,⋆], Matthijs Douze[†], Cordelia Schmid[⋆], and Hervé Jégou[†]

[†]Facebook AI Research          [⋆]Inria

## ABSTRACT

Discretizing multi-dimensional data distributions is a fundamental step of modern indexing methods. State-of-the-art techniques learn parameters of quantizers on training data for optimal performance, thus adapting quantizers to the data. In this work, we propose to reverse this paradigm and adapt the data to the quantizer: we train a neural net which last layer forms a fixed parameter-free quantizer, such as pre-defined points of a hyper-sphere. As a proxy objective, we design and train a neural network that favors uniformity in the spherical latent space, while preserving the neighborhood structure after the mapping. We propose a new regularizer derived from the Kozachenko–Leonenko differential entropy estimator to enforce uniformity and combine it with a locality-aware triplet loss. Experiments show that our end-to-end approach outperforms most learned quantization methods, and is competitive with the state of the art on widely adopted benchmarks. Furthermore, we show that training without the quantization step results in almost no difference in accuracy, but yields a generic catalyzer that can be applied with any subsequent quantizer. The code is available online[1].

## 1 INTRODUCTION

Recent work (Kraska et al., 2017) proposed to leverage the pattern-matching ability of machine learning algorithms to improve traditional index structures such as B-trees or Bloom filters, with encouraging results. In their one-dimensional case, an optimal B-Tree can be constructed if the cumulative density function (CDF) of the indexed value is known, and thus they approximate this CDF using a neural network. We emphasize that the CDF itself is a mapping between the indexed value and a uniform distribution in $[0, 1]$. In this work, we wish to generalize such an approach to multi-dimensional spaces. More precisely, as illustrated by Figure 1, we aim at learning a function that maps real-valued vectors to a uniform distribution over a $d$-dimensional sphere, such that a fixed discretizing structure, for example a fixed binary encoding (sign of components) or a regular lattice quantizer, offers competitive coding performance.

Our approach is evaluated in the context of similarity search, where methods often rely on various forms of learning machinery (Gong et al., 2013; Wang et al., 2014b); in particular there is a substantial body of literature on methods producing compact codes (Jégou et al., 2011a). Yet the problem of jointly optimizing a coding stage and a neural network remains essentially unsolved, partly because

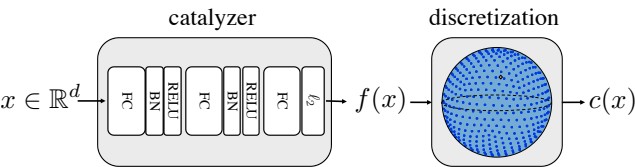

Figure 1: Our method learns a network that encodes the input space $\mathbb{R}^d$ into a code $c(x)$. It is learned end-to-end, yet the part of the network in charge of the discretization operation is fixed in advance, thereby avoiding optimization problems. The learnable function $f$, namely the "catalyzer", is optimized to increase the quality of the subsequent coding stage.

---

[1]https://github.com/facebookresearch/spreadingvectors

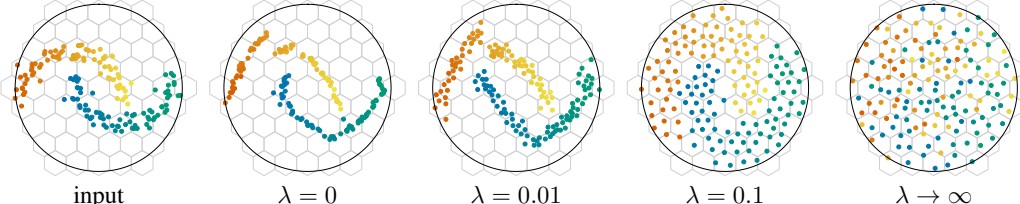

| input | $\lambda = 0$ | $\lambda = 0.01$ | $\lambda = 0.1$ | $\lambda \to \infty$ |

Figure 2: Illustration of our method, which takes as input a set of samples from an unknown distribution. We learn a neural network that aims at preserving the neighborhood structure in the input space while best covering the output space (uniformly). This trade-off is controlled by a parameter $\lambda$. The case $\lambda = 0$ keeps the locality of the neighbors but does not cover the output space. On the opposite, when the loss degenerates to the differential entropic regularizer ($\lambda \to \infty$), the neighbors are not maintained by the mapping. Intermediate values offer different trade-offs between neighbor fidelity and uniformity, which is proper input for an efficient lattice quantizer (depicted here by the hexagonal lattice $A_2$).

it is difficult to optimize through a discretization function. For this reason, most efforts have been devoted to networks producing binary codes, for which optimization tricks exist, such as soft binarization or stochastic relaxation, which are used in conjunction with neural networks (Liong et al., 2015; Jain et al., 2017). However it is difficult to improve over more powerful codes such as those produced by product quantization (Jégou et al., 2011a), and recent solutions addressing product quantization require complex optimization procedures (Klein & Wolf, 2017; Ozan et al., 2016).

In order to circumvent this problem, we propose a drastic simplification of learning algorithms for indexing. We learn a mapping such that the output follows the distribution under which the subsequent discretization method, either binary or a more general quantizer, performs better. In other terms, instead of trying to adapt an indexing structure to the data, we adapt the data to the index.

Our technique requires to jointly optimize two antithetical criteria. First, we need to ensure that neighbors are preserved by the mapping, using a vanilla ranking loss (Usunier et al., 2009; Chechik et al., 2010; Wang et al., 2014a). Second, the training must favor a uniform output. This suggests a regularization similar to maximum entropy (Pereyra et al., 2017), except that in our case we consider a continuous output space. We therefore propose to cast an existing differential entropy estimator into a regularization term, which plays the same "distribution-matching" role as the Kullback-Leiber term of variational auto-encoders (Doersch, 2016).

As a side note, many similarity search methods are implicitly designed for the range search problem (or *near neighbor*, as opposed to *nearest neighbor* (Indyk & Motwani, 1998; Andoni & Indyk, 2006)), that aims at finding all vectors whose distance to the query vector is below a fixed threshold. For real-world high-dimensional data, range search usually returns either no neighbors or too many. The discrepancy between near– and nearest– neighbors is significantly reduced by our technique, see Section 3.3 and Appendix C for details.

Our method is illustrated by Figure 2. We summarize our contributions as follows:

- We introduce an approach for multi-dimensional indexing that maps the input data to an output space in which indexing is easier. It learns a neural network that plays the role of an adapter for subsequent similarity search methods.

- For this purpose we introduce a loss derived from the Kozachenko-Leonenko differential entropy estimator to favor uniformity in the spherical output space.

- Our learned mapping makes it possible to leverage spherical lattice quantizers with competitive quantization properties and efficient algebraic encoding.

- Our ablation study shows that our network can be trained without the quantization layer and used as a plug-in for processing features before using standard quantizers. We show quantitatively that our catalyzer improves performance by a significant margin for quantization-based (OPQ (Ge et al., 2013)) and binary (LSH (Charikar, 2002)) method.

This paper is organized as follows. Section 2 discusses related works. Section 3 introduces our neural network model and the optimization scheme. Section 4 details how we combine this strategy with lattice assignment to produce compact codes. The experimental section 5 evaluates our approach.

## 2    RELATED WORK

**Generative modeling.** Recent models such as Generative Adversarial Networks (GANs) (Goodfellow et al., 2014) or Variational Auto-Encoders (VAEs) (Kingma & Welling, 2013) learn a mapping between an isotropic Gaussian distribution and the empirical distribution of a training set. Our approach maps an empirical input distribution to a uniform distribution on the spherical output space. Another distinction is that GANs learn a unidirectional mapping from the latent code to an image (decoder), whereas VAEs learn a bidirectional mapping (encoder - decoder). In our work, we focus on learning the encoder, whose goal is to pre-process input vectors for subsequent indexing.

**Dimensionality reduction and representation learning.** There is a large body of literature on the topic of dimensionality reduction, see for instance the review by Van Der Maaten et al. (2009). Relevant work includes self-organizing maps (Kohonen et al., 2001), the stochastic neighbor embedding (Hinton & Roweis, 2003) and the subsequent t-SNE approach (van der Maaten & Hinton, 2008), which is tailored to low-dimensional spaces for visualisation purposes. Both works are non-linear dimensionality reduction aiming at preserving the neighborhood in the output space.

**Learning to index and quantize.** The literature on product compact codes for indexing is most relevant to our work, see Wang et al. (2014b; 2016) for an overview of the topic. Early popular high-dimensional approximate neighbor methods, such as Locality Sensitive Hashing (Indyk & Motwani, 1998; Gionis et al., 1999; Charikar, 2002; Andoni & Indyk, 2006), were mostly relying on statistical guarantees without any learning stage. This lack of data adaptation was subsequently addressed by several works. The Iterative quantization (ITQ) (Gong et al., 2013) modifies the coordinate system to improve binarization, while methods inspired by Vector Quantization and compression (Jégou et al., 2011a; Babenko & Lempitsky, 2014; Zhang et al., 2015; Jain et al., 2016) have gradually emerged as strong competitors for estimating distances or similarities with compact codes. While most of these works aim at reproducing target (dis-)similarity, some recent works directly leverage semantic information in a supervised manner with neural networks (Liong et al., 2015; Jain et al., 2017; Klein & Wolf, 2017; Sablayrolles et al., 2017).

**Lattices,** also known as Euclidean networks, are discrete subsets of the Euclidean space that are of particular interest due to their space covering and sphere packing properties (Conway & Sloane, 2013). They also have excellent discretization properties under some assumptions about the distribution, and most interestingly the closest point of a lattice is determined efficiently thanks to algebraic properties (Ran & Snyders, 1998). This is why lattices have been proposed (Andoni & Indyk, 2006; Jégou et al., 2008) as hash functions in LSH. However, for real-world data, lattices waste capacity because they assume that all regions of the space have the same density (Paulevé et al., 2010). In this paper, we are interested in spherical lattices because of their bounded support.

**Entropy regularization** appears in many areas of machine learning and indexing. For instance, Pereyra et al. (2017) argue that penalizing confident output distributions is an effective regularization. Cuturi (2013) use entropy regularization to speed up computation of optimal transport distances. Another proposal by Bojanowski & Joulin (2017) in an unsupervised learning context, is to spread the output by enforcing input images to map to points drawn uniformly on a sphere. Interestingly, most recent works on binary hashing introduce some form of entropic regularization. Deep hashing (Liong et al., 2015) employs a regularization term that increases the marginal entropy of each bit. SUBIC (Jain et al., 2017) extends this idea to one-hot codes.

## 3    OUR APPROACH: LEARNING THE CATALYZER

Our proposal is inspired by prior work for one-dimensional indexing (Kraska et al., 2017). However their approach based on unidimensional density estimation can not be directly translated to the multi-dimensional case. Our strategy is to train a neural network $f$ that maps vectors from a $d_{\text{in}}$-dimensional space to the hypersphere of a $d_{\text{out}}$-dimensional space $\mathcal{S}_{d_{\text{out}}}$.

### 3.1    KoLEO: DIFFERENTIAL ENTROPY REGULARIZER

Let us first introduce our regularizer, which we design to spread out points uniformly across $\mathcal{S}_{d_{\text{out}}}$. With the knowledge of the density of points $p$, we could directly maximize the differential entropy $-\int p(u) \log(p(u)) du$. Given only samples $(f(x_1), ..., f(x_n))$, we instead use an estimator of the

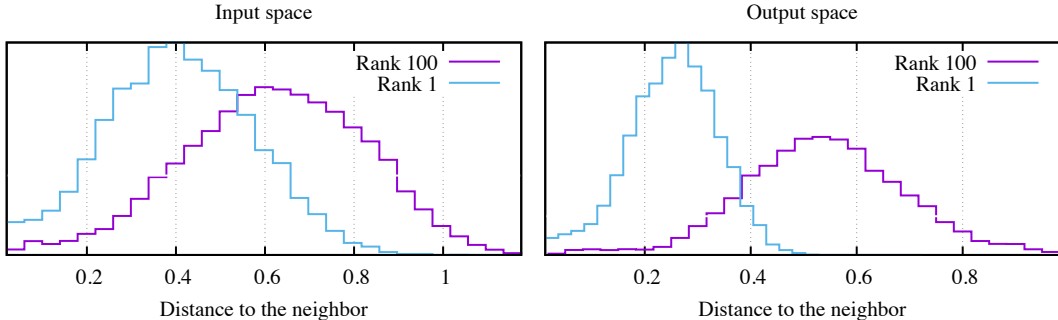

Figure 3: Histograms of the distance between a query point and its 1st (resp. 100$^{\text{th}}$) nearest neighbors, in the original space (left) and after our catalyzer (right). In the original space, the two histograms have a significant overlap, which means that a 100-th nearest neighbor for a query has often a distance lower that the 1st neighbor for another query. This gap is significantly reduced by our catalyzer.

differential entropy as a proxy. It was shown by Kozachenko and Leononenko (see e.g. (Beirlant et al., 1997)) that defining $\rho_{n,i} = \min_{j \neq i} \|f(x_i) - f(x_j)\|$, the differential entropy of the distribution can be estimated by

$$H_n = \frac{\alpha_n}{n} \sum_{i=1}^{n} \log(\rho_{n,i}) + \beta_n, \tag{1}$$

where $\alpha_n$ and $\beta_n$ are two constants that depend on the number of samples $n$ and the dimensionality of the data $d_{\text{out}}$. Ignoring the affine components, we define our entropic regularizer as

$$\mathcal{L}_{\text{KoLeo}} = -\frac{1}{n} \sum_{i=1}^{n} \log(\rho_{n,i}). \tag{2}$$

This loss also has a satisfactory geometric interpretation: closest points are pushed away, with a strength that is non-decreasing and concave. This ensures diminishing returns: as points get away from each other, the marginal impact of increasing the distance becomes smaller.

## 3.2 RANK PRESERVING LOSS

We enforce the outputs of the neural network to follow the same neighborhood structure as in the input space by adopting the triplet loss (Chechik et al., 2010; Wang et al., 2014a)

$$\mathcal{L}_{\text{rank}} = \max \left(0, \|f(x) - f(x^+)\|_2 - \|f(x) - f(x^-)\|_2 \right), \tag{3}$$

where $x$ is a query, $x^+$ a positive match, $x^-$ a negative match. The positive matches are obtained by computing the $k_{\text{pos}}$ nearest neighbors of each point $x$ in the training set in the input space. The negative matches are generated by taking the $k_{\text{neg}}$-th nearest neighbor of $f(x)$ in $(f(x_1), ..., f(x_n))$. In order to speed up the learning, we compute the $k_{\text{neg}}$-th nearest neighbor of every point in the dataset at the beginning of each epoch and use these throughout the epoch. Note that we do not need to use a margin, as its effect is essentially superseded by our regularizer. Our overall loss combines the triplet loss and the entropy regularizer, as

$$\mathcal{L}_{\text{model}} = \mathcal{L}_{\text{rank}} + \lambda \mathcal{L}_{\text{KoLeo}}, \tag{4}$$

where the parameter $\lambda \geq 0$ controls the trade-off between ranking quality and uniformity.

## 3.3 DISCUSSION

**Choice of $\lambda$.** Figure 2 was produced by our method on a toy dataset adapted to the disk as the output space. Without the KoLeo regularization term, neighboring points tend to collapse and most of the output space is not exploited. If we quantize this output with a regular quantizer, many Voronoi cells are empty and we waste coding capacity. In contrast, if we solely rely on the entropic regularizer, the neighbors are poorly preserved. Interesting trade-offs are achieved with intermediate values of $\lambda$.

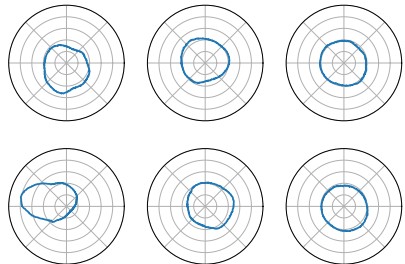

Figure 4: Impact of the regularizer on the output distribution. Each column corresponds to a different amount of regularization (left: $\lambda = 0$, middle: $\lambda = 0.02$, right: $\lambda = 1$). Each line corresponds to a different random projection of the empirical distribution, parametrized by an angle in $[0, 2\pi]$. The marginal distributions for these two views are much more uniform with our KoLeo regularizer, which is a consequence of the higher uniformity in the high-dimensional latent space.

**Qualitative evaluation of the uniformity.** Figure 3 shows the histogram of the distance to the nearest (resp. 100th nearest) neighbor, before applying the catalyzer (left) and after (right). The overlap between the two distributions is significantly reduced by the catalyzer. We evaluate this quantitatively by measuring the probability that the distance between a point and its nearest neighbor is larger than the distance between another point and its 100th nearest neighbor. In a very imbalanced space, this value is 50%, whereas in a uniform space it should approach 0%. In the input space, this probability is 20.8%, and it goes down to 5.0% in the output space thanks to our catalyzer.

**Visualization of the output distribution.** While Figure 2 illustrates our method with the 2D disk as an output space, we are interested in mapping input samples to a higher dimensional hyper-sphere. Figure 4 proposes a visualization of the high-dimensional density from a different viewpoint, with the Deep1M dataset mapped in 8 dimensions. We sample 2 planes randomly in $\mathbb{R}^{d_{\text{out}}}$ and project the dataset points $(f(x_1), ..., f(x_n))$ on them. For each column, the 2 figures are the angular histograms of the points with a polar parametrization of this plane. The area inside the curve is constant and proportional to the number of samples $n$. A uniform angular distribution produces a centered disk, and less uniform distributions look like unbalanced potatoes.

The densities we represent are marginalized, so if the distribution looks non-uniform then it is non-uniform in $d_{\text{out}}$-dimensional space, but the reverse is not true. Yet one can compare the results obtained for different regularization coefficients, which shows that our regularizer has a strong uniformizing effect on the mapping, ultimately resembling that of a uniform distribution for $\lambda = 1$.

## 4 CATALYZER WITH DISCRETIZATION

In this section we describe how our method interplays with discretization, at training and at search time. We consider two parameter-free coding methods: binarization and defining a fixed set of points on the unit sphere provided by a lattice spherical quantizer. A key advantage of a fixed coding structure like ours is that compressed-domain distance computations between codes do not depend on external meta-data. This is in contrast with quantization-based methods like product quantization, which require centroids to be available at search time.

### 4.1 BINARIZATION

Binary features are obtained by applying the `sign` function to the coordinates. We relax this constraint at train time by replacing the `sign` with the identity function, and the binarization is used only to cross-validate the regularization parameter on the validation set.

### 4.2 LATTICES

As discussed by Paulevé et al. (2010), lattices impose a rigid partitioning of the feature space, which is suboptimal for arbitrary distributions, see Figure 2. In contrast, lattices offer excellent quantization properties for a uniform distribution (Conway & Sloane, 2013). Thanks to our regularizer, we are closer to uniformity in the output space, making lattices an attractive choice.

We consider the simplest spherical lattice, integer points of norm $r$, a set we denote $S_d^r$. Given a vector $x \in \mathbb{R}^{d_{\text{in}}}$, we compute its catalyzed features $f(x)$, and find the nearest lattice point on $S_d^r$ using the assignment operation, which formally minimizes $q(f(x)) = \min_{c \in S_d^r} \| r \times f(x) - c \|_2^2$.

This assignment can be computed very efficiently (see Appendix B for details). Given a query $y$ and its representation $f(y)$, we approximate the similarity between $y$ and $x$ using the code: $\|f(y) - f(x)\|_2 \approx \|f(y) - q(f(x))/r\|_2$, This is an asymmetric comparison, because the query vectors are not quantized (Jégou et al., 2011a).

When used as a layer, it takes a vector in $\mathbb{R}^d$ and returns the quantized version of this vector in the forward pass, and passes the gradient to the previous layer in the backward pass. This heuristic is referred to as the *straight-through* estimator in the literature, and is often used for discretization steps, see *e.g.*, van den Oord et al. (2017).

## 5 EXPERIMENTS

This section presents our experimental results. We focus on the class of similarity search methods that represents the database vectors with a compressed representation (Charikar, 2002; Jégou et al., 2011a; Gong et al., 2013; Ge et al., 2013), which enables to store very large dataset in memory (Lv et al., 2004; Torralba et al., 2008).

### 5.1 EXPERIMENTAL SETUP

All experiments have two phases. In the first phase (encoding), all vectors of a database are encoded into a representation (*e.g.* 32, 64 bits). Encoding consists in a vector transformation followed by a quantization or binarization stage. The second phase is the search phase: a set of query vectors is transformed, then the codes are scanned exhaustively and compared with the transformed query vector, and the top-k nearest vectors are returned.

**Datasets and metrics.** We use two benchmark datasets Deep1M and BigAnn1M. Deep1M consists of the first million vectors of the Deep1B dataset (Babenko & Lempitsky, 2016). The vectors were obtained by running a convnet on an image collection, reduced to 96 dimensions by principal component analysis and subsequently $\ell_2$-normalized. We also experiment with the BigAnn1M (Jégou et al., 2011b), which consists of SIFT descriptors (Lowe, 2004). Both datasets contain 1M vectors that serve as a reference set, 10k query vectors and a very large training set of which we use 500k elements for training, and 1M vectors that we use a base to cross-validate the hyperparameters $d_{\text{out}}$ and $\lambda$. We also experiment on the full Deep1B and BigAnn datasets, that contain 1 billion elements. We evaluate methods with the recall at $k$ performance measure, which is the proportion of results that contain the ground truth nearest neighbor when returning the top $k$ candidates (for $k \in \{1, 10, 100\}$).

**Training.** For all methods, we train our neural network on the training set, cross-validate $d_{\text{out}}$ and $\lambda$ on the validation set, and use a different set of vectors for evaluation. In contrast, some works carry out training on the database vectors themselves (Muja & Lowe, 2014; Malkov & Yashunin, 2016; Gong et al., 2013), in which case the index is tailored to a particular fixed set of database vectors.

### 5.2 MODEL ARCHITECTURE AND OPTIMIZATION

Our model is a 3 - layer perceptron, with ReLU non-linearity and hidden dimension $1024$. The final linear layer projects the dataset to the desired output dimension $d_{\text{out}}$, along with $\ell_2$-normalization. We use batch normalization (Ioffe & Szegedy, 2015) and train our model for 300 epochs with Stochastic Gradient Descent, with an initial learning rate of $0.1$ and a momentum of $0.9$. The learning rate is decayed to $0.05$ (resp. $0.01$) at the 80-th epoch (resp. 120-th).

### 5.3 SIMILARITY SEARCH WITH LATTICE VECTOR QUANTIZERS

We evaluate the lattice-based indexing proposed in Section 4, and compare it to more conventional methods based on quantization, namely PQ (Jégou et al., 2011a) and Optimized Product Quantization (OPQ) (Ge et al., 2013). We use the Faiss (Johnson et al., 2017) implementation of PQ and OPQ and assign one byte per sub-vector (each individual quantizer has 256 centroids). For our lattice, we vary the value of $r$ to increase the quantizer size, hence generating curves for each value of $d_{\text{out}}$. Figure 5 provides a comparison of these methods. On both datasets, the lattice quantizer strongly outperforms PQ and OPQ for most code sizes.

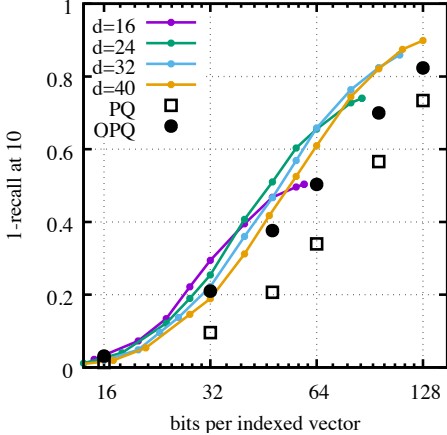 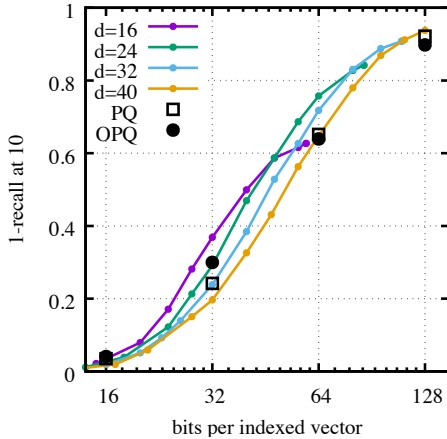

Figure 5: Comparison of the performance of the product lattice *vs* OPQ on Deep1M (left) and BigAnn1M (right). Our method maps the input vectors to a $d_{\text{out}}$-dimensional space, that is then quantized with a lattice of radius $r$. We obtain the curves by varying the radius $r$.

**Impact of the hyperparameters.** Varying the rank parameters $k_{\text{pos}}$ and $k_{\text{neg}}$ did not impact significantly the performance, so we fixed them respectively to $k_{\text{pos}} = 10$ and $k_{\text{neg}} = 50$. For a fixed number of bits, varying the dimension $d_{\text{out}}$ is a trade-off between a good representation and an easily compressible one. When $d_{\text{out}}$ is small, we can use a large $r$ for a very small quantization error, but there are not enough dimensions to represent the degrees of freedom of the underlying data. A larger $d_{\text{out}}$ allows for better representations but suffers from a coarser approximation. Figure 5 shows that for low bitrates, small dimensions perform better because the approximation quality dominates, whereas for higher bitrates, larger dimensions are better because the representation quality dominates. Similarly, the regularizer $\lambda$ needs to be set to a large value for small dimensions and low bitrates, but higher dimensions and higher bitrates require lower values of $\lambda$ (cf. Appendix A for details).

**Large-scale experiments.** We experiment with the full Deep1B (resp. BigAnn) dataset, that contains 1 billion vectors, with 64 bits codes. At that scale, the recall at 10 drops to 26.1% for OPQ and to 37.8% for the lattice quantizer (resp. 21.3% and 36.5%). As expected, the recall performance is lower than for the 1 million vectors database, but the precision advantage of the lattice quantizer is maintained at large scale.

**Comparison to the state of the art.** Additive quantization variants (Babenko & Lempitsky, 2014; Martinez et al., 2018; Ozan et al., 2016) are currently state-of-the art encodings for vectors in terms of accuracy. However, their encoding stage involves an iterative optimization process that is prohibitively slow for practical use cases. For example, Competitive quantization's reported complexity is $15\times$

| Recall1@ | Deep1M | | | BigAnn1M | | | Encoding time |
|---|---|---|---|---|---|---|---|
| | 1 | 10 | 100 | 1 | 10 | 100 | (1M vectors) |
| OPQ (Ge et al., 2013) | 15.6 | 50.3 | 88.1 | 20.8 | 63.6 | 95.3 | 5.5 s |
| Catalyst + OPQ | 21.2 | 62.8 | 93.4 | 24.9 | 71.1 | 97.0 | 11.4 s |
| LSQ (Martinez et al., 2018) | 20.3 | 61.4 | 94.3 | 28.4 | **76.2** | **98.7** | 122.1 s |
| PCA + Lattice | 12.2 | 42.5 | 81.6 | 19.0 | 60.6 | 93.5 | 5.3 s |
| Catalyst + Lattice | 22.6 | 66.9 | 95.2 | 28.4 | 75.8 | 98.3 | 8.5 s |
| Catalyst + Lattice + end2end | **22.8** | **67.5** | **95.5** | **28.7** | **76.2** | 98.2 | 8.5 s |

Table 1: Comparison of different flavors of the catalyst: with a lattice quantizer (with or without end-to-end training), and with OPQ. All results use 64 bits per code. All timings are for BigAnn1M are on a 2.2 GHz machine with 40 threads. The encoding times associated with the catalyzer include the forward pass through our neural network. Note, our lattice-based coding scheme is the only one not requiring external meta-data once the compact code is produced.

|  | Deep1M | | | | BigAnn1M | | | |
| --- | --- | --- | --- | --- | --- | --- | --- | --- |
| bits per vector | 16 | 32 | 64 | 128 | 16 | 32 | 64 | 128 |
| LSH | 0.8 | 4.9 | 14.6 | 32.2 | 0.3 | 2.6 | 9.5 | 24.4 |
| ITQ | 1.0 | 7.3 | 21.0 | n/a | 1.5 | 8.5 | 22.8 | 41.8 |
| Catalyzer + `sign` | **2.7** | **11.4** | **25.5** | **46.8** | **2.6** | **11.1** | **28.3** | **52.2** |

Table 2: Performance (1-recall at 10, %) with LSH, on Deep1M and BigAnn1M, as a function of the number of bits per index vector. All results are averaged over 5 runs with different random seeds. Our catalyzer gets a large improvement in binary codes over LSH and ITQ.

slower than OPQ. Table 1 compares our results with LSQ (Martinez et al., 2018), a recent variant that is close to the state of the art and for which open-source code is available. We show that our Catalyst + Lattice variant method is $14\times$ times faster for an accuracy that is competitive or well above that of LSQ. To our knowledge, this is the first time that such competitive results are reported for a method that can be used in practice at a large scale. Our search time is a bit slower: computing 1M asymmetric distances takes 7.5 ms with the Catalyzer+Lattice instead of 4.9 ms with PQ. This is due to our decoding procedure, which does not rely on precomputed tables as used in PQ.

## 5.4 A UNIVERSAL CATALYZER?

**Ablation study.** As a sanity check, we first replace our catalyzer by a PCA that reduces the dimensionality to the same size as our catalyzer, followed by $\ell_2$-normalization. This significantly decreases the performance of the lattice quantizer, as can be seen in Table 1.

We also evaluate the impact of training end-to-end, compared to training without the quantization layer. Table 1 shows that end-to-end training has a limited impact on the overall performance for $64$ bits, sometimes even decreasing performance. This may be partly due to the approximation induced by the straight-through estimation, which handicaps end-to-end training. Another reason is that the KoLeo regularizer narrows the performance gap induced by discretization. In other terms, our method trained without the discretization layer trains a general-purpose network (hence the name *catalyzer*), on which we can apply any binarization or quantization method. Table 1 shows that OPQ is improved when applied on top of catalyzed features, for example increasing the recall@10 from 63.6 to 71.1.

**Binary hashing.** We also show the interest of our method as a catalyzer for binary hashing, compared to two popular methods (Charikar, 2002; Gong et al., 2013):

**LSH** maps Euclidean vectors to binary codes that are then compared with Hamming distance. A set of $m$ fixed projection directions are drawn randomly and isotropically in $d_{in}$, and each vector is encoded into $m$ bits by taking the sign of the dot product with each direction.

**ITQ** is another popular hashing method, that improves LSH by using an orthogonal projection that is optimized to maximize correlation between the original vectors and the bits.

Table 2 compares our catalyzer to LSH and ITQ. Note that a simple `sign` function is applied to the catalyzed features. The catalyzer improves the performance by 2-9 percentage points in all settings, from 32 to 128 bits.

## 6 CONCLUDING REMARKS

We train a neural network that maps input features to a uniform output distribution on a unit hypersphere, making high-dimensional indexing more accurate, in particular with fast and rigid lattice quantizers or a trivial binary encoding. To the best of our knowledge, this is the first work on multi-dimensional data that demonstrates that it is competitive to adapt the data distribution to a rigid quantizer, instead of adapting the quantizer to the input data. This has several benefits: rigid quantizers are fast at encoding time; and vectors can be decoded without carrying around codebooks or auxiliary tables. We open-sourced the code corresponding to the experiments at `https://github.com/facebookresearch/spreadingvectors`.

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

## APPENDIX A   VALUES OF THE REGULARIZATION PARAMETER

The optimal value of the regularizer $\lambda$ decreases with the dimension, as shown by Table 3.

| $d_{\text{out}}$ | $\lambda$ |
|---|---|
| 16 | 0.05 |
| 24 | 0.02 |
| 32 | 0.01 |
| 40 | 0.005 |

Table 3: Optimal values of the regularization parameter $\lambda$ for Deep1M, using a fixed radius of $r = 10$.

## APPENDIX B   FAST DISCRETIZATION WITH A LATTICE ON THE SPHERE.

We consider the set of integer points $z = (z_1, ..., z_d) \in \mathbb{Z}^d$ such that $\sum_{i=1}^d z_i^2 = r^2$, that we denote $S_d^r$. This set is the intersection of the hyper-cubic lattice $\mathbb{Z}^d$ with the hyper-sphere of radius $r$. For example to extract a $64-$bit representation in 24D we use $r^2 = 79$. Quantizing a vector $y \in \mathbb{R}^d$ amounts to solving the following optimization problem:

$$\underset{z \in S_d^r}{\operatorname{argmin}} \|y - z\|^2 = \underset{z \in S_d^r}{\operatorname{argmax}} \ yz^\top. \tag{5}$$

**Atoms.**   We define a "normalization" function $N$ of vectors: it consists in taking the absolute value of their coordinates, and sorting them by decreasing coordinates. We call "atoms" the set of vectors that can be obtained by normalizing the vectors of $S_d^r$.

For example, the atoms of $S_8^{\sqrt{10}}$ are:

$$\begin{cases} 3 & 1 & 0 & 0 & 0 & 0 & 0 & 0 \\ 2 & 2 & 1 & 1 & 0 & 0 & 0 & 0 \\ 2 & 1 & 1 & 1 & 1 & 1 & 1 & 0 \end{cases} \tag{6}$$

All vectors of $S_d^r$ can be represented as permutations of an atom, with sign flips. Figure 6 reports the number of vectors of $S_d^k$ and the corresponding number of atoms.

**Encoding and enumerating.**   To solve Equation 5, we apply the following steps:

1. normalize $y$ with $N$, store the permutation $\sigma$ that sorts coordinates of $|y|$
2. exhaustively search the atom $z'$ that maximizes $N(y)^\top z'$
3. apply the inverse permutation $\sigma^{-1}$ that sorts $y$ to $z'$ to obtain $z''$
4. the nearest vector $(z_1, .., z_d)$ is $z_i = \operatorname{sign}(y_i) z_i'' \quad \forall i = 1..d$.

To encode a vector of $z \in S_d^r$ we proceed from $N(z)$:

1. each atom is assigned a range of codes, so $z$ is encoded relative to the start of $N(z)$'s range
2. encode the permutation using combinatorial number systems Knuth (2005). There are $d!$ permutations, but the permutation of equal components is irrelevant, which divides the number combinations. For example atom $(2, 2, 1, 1, 0, 0, 0, 0)$ is the normalized form of $8!/(2!2!4!) = 240$ vectors of $S_8^{\sqrt{10}}$.
3. encode the sign of non-zero elements. In the example above, there are 4 sign bits.

Decoding proceeds in the reverse order.

Encoding 1M vectors takes about 0.5 s on our reference machine, which is faster than PQ (1.9 s). In other terms, he quantization time is negligible w.r.t. the preprocessing by the catalyzer.

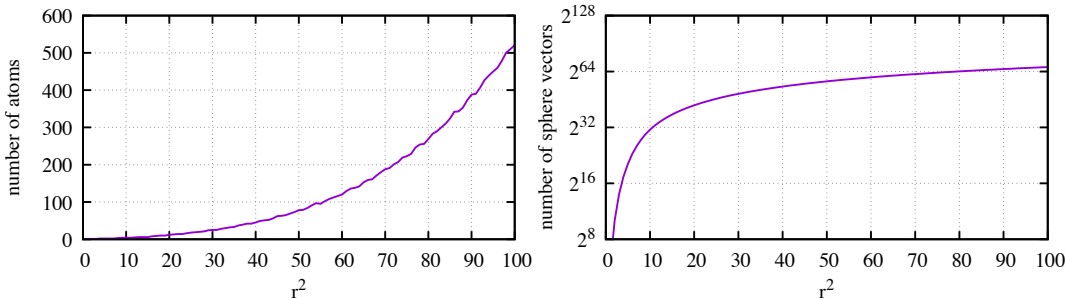

Figure 6: Number of atoms of the hyper-sphere of $\mathcal{S}_{24}^{r}$. (linear scale), and the corresponding number of points on the hyper-sphere (log scale).

## APPENDIX C    EPSILON-SEARCH

Figure 7 shows how our method achieves a better agreement between range search and k-nearest neighbors search on Deep1M. In this experiment, we consider different thresholds $\varepsilon$ for the range search and perform a set of queries for each $\varepsilon$. Then we measure how many vectors we must return, on average, to achieve a certain recall in terms of the nearest neighbors in the original space. Without our mapping, there is a large variance on the number of results for a given $\varepsilon$. In contrast, after the mapping it is possible to use a unique threshold to find most neighbors.

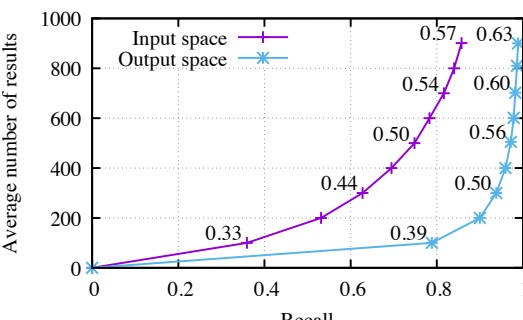

Figure 7: Agreement between nearest neighbor and range search: average number of results per query for given values of $\varepsilon$ (indicated on the curve), and corresponding recall values. For example: to obtain 80% recall, the search in the original space requires to set $\varepsilon = 0.54$, which returns 700 results per query on average, while in the transformed space $\varepsilon = 0.38$ returns just 200 results. Observe the much better agreement in the latent spherical space.

