# OpenReview forum: "Spreading vectors for similarity search"
_ICLR.cc/2019/Conference_

### Official Review · AnonReviewer1 · 2018-10-20
**Problematic  experimental results**

**Rating:** 6
**Confidence:** 4

**Review:**

The idea, transforming the input data to an output space in which the data is distributed uniformly and thus indexing is easier, is interesting.

My main concerns come from experimental results.

(1) Table 1: where are the results of OPQ and LSQ from? run the codes by the authors of this paper? or from the original paper?

It is not consistent to the LSQ paper (https://www.cs.ubc.ca/~julm/papers/eccv16.pdf). For BigANN1M, from the LSQ paper, the result is >29 recall at 1 for 64 bits.

(2) Figure 5: similarly, how did you get the results of PQ and OPQ?

(3) There are some other advanced algorithms: e.g.,  additive quantization (Babenko & Lempitsky, 2014) and composite quantization (https://arxiv.org/abs/1712.00955)

The above points make it hard to judge this paper.

---

> ### Author Response · Authors · 2018-11-13
> **Response to Review #1**
>
> We thank the reviewer for their review and comments. We provide detailed answers below.
>
> "My main concerns come from experimental results."
> Upon publication of the paper, we will release the code that replicates the experiments.
>
> "(1) Table 1: where are the results of OPQ and LSQ from?  [...] It is not consist to the LSQ paper"
> In our experiments, we used the reference public implementation from the authors of LSQ [1]. The discrepancy in the reported 64-bit recall1@1 comes from the fact that the datasets are different: we use Bigann1m (28.4 recall) whereas the LSQ paper reports results on Sift1m. We conducted experiments on Bigann1m because the training set associated with Sift1m is too small (100k vectors) for learning the catalyzer. As a sanity check, we re-ran the code of [1] on Sift1m and obtained 28.99, which is consistent with the results reported by [A] (Table 1, LSQ, 29.37) and [Martinez et al, 2018] (Figure 3, LSQ SR-C and SR-D, ~28; Table 4 corresponds to another experimental setting).
>
> “(2) Figure 5: similarly, how did you get the results of PQ and OPQ?”
> We used the open-source Faiss library [2] to obtain the results of PQ and OPQ. This library is used as a reference implementation in recent papers like [D, E]. There is a comparison point with [F] on Deep1M at 64 bit: the Faiss implementation of OPQ obtains recall@1 = 15.6 vs 16.1 in [F] (table 1).
>
> “(3) There are some other advanced algorithms: e.g., additive quantization [B] and composite quantization [C]”
> We did not compare directly to AQ and CQ as it was shown that they underperform LSQ by some margin (Table 1 in [A]). Besides, in general, we insist in the paper that the encoding time for additive quantization methods is at least an order of magnitude slower than product quantization and our catalyzer + lattice (122s for LSQ vs < 10s for PQ/Lattice, cf. Table 1).
>
> References
> [1] https://github.com/una-dinosauria/local-search-quantization
> [2] https://github.com/facebookresearch/faiss
>
> [A] Revisiting additive quantization, Martinez et al., ECCV'2016
> [B] Additive Quantization for Extreme Vector Compression, Babenko & Lempitsky, CVPR'2014
> [C] Composite Quantization, J Wang et al. ICML'14
> [D] Link and code: Fast indexing with graphs and compact regression codes, Douze et al, CVPR'18
> [E] Revisiting the Inverted Indices for Billion-Scale Approximate Nearest Neighbors, Baranchuk et al, ECCV'18
> [F] AnnArbor: Approximate Nearest Neighbors Using Arborescence Coding, Babenko & Lempitsky, ICCV'17

---

### Official Review · AnonReviewer3 · 2018-10-28
**Well motivated novel idea; excellent results**

**Rating:** 7
**Confidence:** 4

**Review:**

Pros
----

[Originality]
The authors propose a novel idea of learning representations that improves the performance of the subsequent fixed discretization method.

[Clarity]
The authors clearly motivate their solution and explain the different ideas and enhancements introduced. The manuscript is fairly easy to follow. The different terms in the optimization problem are clearly explained and their individual behaviour are presented for the better understanding.

[Significance]
The empirical results for the proposed scheme are compared against various baselines under various scenarios and the results demonstrate the significant utility of the proposed scheme.

Limitations
-----------

[Clarity]
The training times for the catalyzer is never discussed in this manuscript (even relative to the training times of the considered baselines). Moreover, it is not clear if the inference time of the catalyzer is included in the results such as Table 1. Even if, PQ and the catalyzer+lattice might have comparable search recalls, it would be good to understand the relative search times to get similar accuracy especially since the inference time for the catalyzer (which is part of the search time) can be fairly significant.

[Clarity/Significance]
One important point not discussed in this manuscript is the choice of the structure (architechture) of the catalyzer. Is the catalyzer architecture dependent on the data?
  - If yes, how to find an appropriate architecture?
  - If no, what is it about the proposed architecture that makes it sufficient for all data sets?
In my opinion, this is extremely important since this drives the applicability of the proposed scheme beyond the presented examples.

[Minor question]
- Is the parameter r in the rank loss same as the norm r in the lattice quantizer? This is a bit confusing.

---

> ### Author Response · Authors · 2018-11-13
> **Response to Review #3**
>
> We thank the reviewer for their comments.
>
> “The training times for the catalyzer is never discussed in this manuscript. [...] Moreover, it is not clear if the inference time of the catalyzer is included in the results such as Table 1.”
> Training takes between 2 and 3 hours on a CPU machine using 20 cores, and the reported query timings take into account inference.
>
> “One important point not discussed in this manuscript is the choice of the structure (architecture) of the catalyzer. Is the catalyzer architecture dependent on the data?”
> Generally, we observe that beyond 3 layers there is no improvement in accuracy. The performance improves when augmenting the width of the network, but with diminishing returns. We use the same architecture across datasets. We successfully used the same architecture on other datasets, but we report results here on the standard datasets of the field.
>
> “What is it about the proposed architecture that makes it sufficient for all data sets?”
> We have observed that the dimensions of the hidden layers in our architecture provide enough representation power for the model to be performant across all the datasets we have tested (those of the paper plus some internal datasets).
>
> “Is the parameter r in the rank loss same as the norm r in the lattice quantizer?”
> The parameter r is not the same as the norm r of the lattice quantizer, we thank the reviewer for spotting this, we will update the paper to lift this ambiguity.

---

> > ### Comment · AnonReviewer3 · 2018-11-25
> > **Thank you for the response**
> >
> > Thank you for your response regarding the training times. However, the response regarding the network architecture leaves me unsatisfied. The response (mostly) tells me how the proposed architecture works. It does not really tell me how the authors landed on this architecture. Without knowing how we got here, it is hard to know if/how we can improve it.
> >
> > Also, related to 'The performance improves when augmenting the width of the network, but with diminishing returns.', how was the width of the network(s) used for the empirical evaluation selected. Were they fixed to some value (if so, why) or were they cross-validated over?

---

> > > ### Author Response · Authors · 2018-11-27
> > > **The architecture was cross-validated**
> > >
> > > Thank you for your feedback.
> > >
> > > 1) Regarding the architecture, we discarded many choices that are tailored to specific assumptions (like convolutional layers that assume spatial or temporal shift-invariance), since we do not make these assumptions about our data. Then, we started from a simple multi-layer perceptron, with the standard ReLU non-linearity and pre-activation batch norm. We chose to keep the same width at each layer, and varied depth and width to find optimal parameters (see details below).
> > > We also tried several variants of the architecture and the training regime that did not improve performance or made it worse:
> > >
> > > * other types of architectures  (we tested expansion: 128-256-512, 512-1024; reduction: 64-48-32, 1024-512; and both: 128-256-512-256-128-64)
> > > * sampling new training points every nth epoch (tried every epoch, every 3, 10, 30 epochs)
> > > * varying learning rate decay (factors 1, 1.01, 1.05 or 1.1)
> > > * variants of hard negative mining: only using negatives for which there is no positive closer to the query point
> > >
> > > None of these variants improved above the architecture/schedule, which had the advantage of being conceptually simpler.
> > > The parameters reported in the paper were the best we found with the amount of training data available at hand and with our standard optimization scheme. To conclude on this point, on one side it is difficult to answer the question "if/how we can improve it" for most works in deep learning. What we suggested for the architecture is the best choice we found to date. On the other side, from our experiments, we generally observed that the choice of the discretization layer has more impact on the overall performance of the system than the choice of a particular structure for the catalyzer (compare binary and lattice for instance), therefore there is likely some margin on improvement in this direction.
> > >
> > > 2) "The performance improves when augmenting the width of the network, but with diminishing returns.', how was the width of the network(s) used for the empirical evaluation selected. Were they fixed to some value (if so, why) or were they cross-validated over?"
> > >
> > > We conducted preliminary experiments on the validation set, varying the depth of the network from 0 to 6 latent layers, and the size of the latent layers from 128 to 2048 by doubling it every time. The best results with respect to depth were obtained with 2 latent layers; doubling the width provided increments of 2 - 5 points until 512, then 512 → 1024 resulted in a 0.5 improvement and 1024→ 2048 resulted in a negligible improvement while having a longer runtime.

---

> > > > ### Comment · AnonReviewer3 · 2018-11-29
> > > > **Thank you for the detailed explanation**
> > > >
> > > > The above explanation has been very helpful and would be a great addendum to the manuscript.
> > > >
> > > > Minor question: In table 1, what is 'Catalyst + Lattice + end2end' and how is it different from 'Catalyst + Lattice'? I have been unable to find an explanation anywhere in the manuscript.

---

> > > > > ### Author Response · Authors · 2018-12-03
> > > > > **End2end uses a quantization layer at training time**
> > > > >
> > > > > Thank you for the feedback. The method "Catalyst + Lattice + end2end" refers to using a quantization layer during training with the straight-through estimator described in Section 4.2. In contrast, the version "Catalyst + Lattice" also optimizes Eqn (4) but without including the quantization layer during training. We will clarify this point in the manuscript, and add the precisions regarding the architecture as suggested above.

---

### Official Review · AnonReviewer2 · 2018-10-31
**Spreading vectors for similarity search**

**Rating:** 6
**Confidence:** 3

**Review:**

The authors propose a method to adapt the data to the quantizer, instead of having to work with a difficult to optimize discretization function. The contribution is interesting.

Additional comments and suggestions:

- in the related work overview it would be good to also check possible connections with optimal transport methods using entropy regularization.

- at some points in the paper, e.g. section 3.3, the authors mention Voronoi cells. However, in the related work in section 2 vector quantization and self-organizing maps have not been mentioned.

- more details on the optimization or learning algorithms for eq (3)(4) should be given. The loss function is non-smooth and rather complicated. What are the implications on the learning algorithm when training neural networks? Is it important to have a good initialization or not?

- How reproducible are the results? In Table 1 only one number in each column is shown while eqs (3)(4) are non-convex problems. Is it the best result of several runs or an average that is reported in the Table?

---

> ### Author Response · Authors · 2018-11-13
> **Response to Review #2**
>
> We thank the reviewer for their review. Upon publication of the paper, we will open-source the code that replicates the experiments. In the meantime, we provide more details below:
>
> 1) “in the related work overview it would be good to also check possible connections with optimal transport methods using entropy regularization.“
> In the related work, we mention Bojanowski & Joulin (2017), who use optimal transport (without entropy regularization) to match images with random points on the sphere. The entropy regularization in optimal transport is a bit different than our entropy regularization as it is used mainly for speed purposes, whereas our entropy regularization provides a trade-off between the quality of nearest neighbors and how spread-out the output of the neural network is.
>
> 2) “at some points in the paper, e.g. section 3.3, the authors mention Voronoi cells. However, in the related work in section 2 vector quantization and self-organizing maps have not been mentioned.”
> We will update the related work with these references. In the context of section 3.3, Voronoi cells correspond to the quantization cells of the lattice.
>
> 3) “more details on the optimization or learning algorithms for eq (3)(4) should be given. The loss function is non-smooth and rather complicated. What are the implications on the learning algorithm when training neural networks? Is it important to have a good initialization or not?”
> We found that standard practice for training neural networks worked quite well in our setting (even though we have no guarantee of getting to the global minimum of the objective function). More specifically, we train our networks with Stochastic Gradient descent with an initial learning rate of 0.5, momentum of 0.9, and decay the learning rate when the validation accuracy does not go up for an epoch. We did not need specific initialization to make the networks converge.
>
> 4) “How reproducible are the results? In Table 1 only one number in each column is shown while eqs (3)(4) are non-convex problems. Is it the best result of several runs or an average that is reported in the Table? “
> Our preliminary experiments have shown that the difference in performance between different trainings is very small (despite the problems being non-convex).  Therefore we train only once per set of hyper-parameters (d_out and lambda), and report the corresponding result. Our open-source code will reproduce these results up to the (very small) variations due to random initialization and mini-batch sampling.

---

### Author Response · Authors · 2018-11-23
**Updated paper**

We updated the paper to correct typos, add some precisions and include the references suggested by the reviewers.

---

### Meta-Review · Area_Chair1 · 2018-12-13
**novel and effective method for more effective quantisation of vectorial representations**

**Confidence:** 4
**Recommendation:** Accept (Poster)

**Metareview:**

. Describe the strengths of the paper.  As pointed out by the reviewers and based on your expert opinion.

- The proposed method is novel and effective
- The paper is clear and the experiments and literature review are sufficient (especially after revision).

2. Describe the weaknesses of the paper. As pointed out by the reviewers and based on your expert opinion. Be sure to indicate which weaknesses are seen as salient for the decision (i.e., potential critical flaws), as opposed to weaknesses that the authors can likely fix in a revision.

The original weaknesses (mainly clarity and missing details) were adequately addressed in the revisions.

3. Discuss any major points of contention. As raised by the authors or reviewers in the discussion, and how these might have influenced the decision. If the authors provide a rebuttal to a potential reviewer concern, it’s a good idea to acknowledge this and note whether it influenced the final decision or not. This makes sure that author responses are addressed adequately.

No major points of contention.

4. If consensus was reached, say so. Otherwise, explain what the source of reviewer disagreement was and why the decision on the paper aligns with one set of reviewers or another.

The reviewers reached a consensus that the paper should be accepted.